# Discovery of novel determinants of endothelial lineage using chimeric heterokaryons

**Wing Tak Wong[1†], Gianfranco Matrone[1†], XiaoYu Tian[1], Simion Alin Tomoiaga[1], Kin Fai Au[1,2], Shu Meng[1], Sayumi Yamazoe[3], Daniel Sieveking[3], Kaifu Chen[1], David M Burns[4], James K Chen[3], Helen M Blau[4], John P Cooke[1*]**

[1]Department of Cardiovascular Sciences, Houston Methodist Research Institute, Houston, United States; [2]Department of Internal Medicine, University of Iowa, Iowa City, United States; [3]Department of Chemical and Systems Biology, Stanford University School of Medicine, Stanford, United States; [4]Baxter Laboratory for Stem Cell Biology, Stanford University School of Medicine, Stanford, United States

**Abstract** We wish to identify determinants of endothelial lineage. Murine embryonic stem cells (mESC) were fused with human endothelial cells in stable, non-dividing, heterokaryons. Using RNA-seq, it is possible to discriminate between human and mouse transcripts in these chimeric heterokaryons. We observed a temporal pattern of gene expression in the ESCs of the heterokaryons that recapitulated ontogeny, with early mesodermal factors being expressed before mature endothelial genes. A set of transcriptional factors not known to be involved in endothelial development was upregulated, one of which was POU class 3 homeobox 2 (Pou3f2). We confirmed its importance in differentiation to endothelial lineage via loss- and gain-of-function (LOF and GOF). Its role in vascular development was validated in zebrafish embryos using morpholino oligonucleotides. These studies provide a systematic and mechanistic approach for identifying key regulators in directed differentiation of pluripotent stem cells to somatic cell lineages.

*For correspondence: jpcooke@houstonmethodist.org

†These authors contributed equally to this work

Competing interests: The authors declare that no competing interests exist.

## Introduction

Our understanding of the genetic and epigenetic processes governing endothelial development and differentiation is limited (*Yan et al., 2010*; *De Val and Black, 2009*). Accordingly, our methodologies for obtaining endothelial cells from pluripotent stem cells are empirically driven and suboptimal (*Choi et al., 2009*; *James et al., 2010*; *Huang et al., 2010a*, *2010b*; *Wong et al., 2012*). There is unexplained inconsistency in the yield of iPSC-ECs; in the stability of their phenotype; and in the fidelity of differentiation (in terms of replicating the epigenetic and genetic profile of a mature endothelial cell). Furthermore, our ability to efficiently generate specific endothelial subtypes (e.g. arterial, venous, lymphatic) is poor. Thus, a systematic approach is needed to more completely define the genetic and epigenetic programs required for differentiating pluripotent stem cells to the endothelial phenotype. Here, we propose an unbiased systematic approach to discover determinants of differentiation. We use interspecies heterokaryons, RNA sequencing and third-generation bioinformatics to discover novel candidate genes critical for proper endothelial differentiation and specification.

**eLife digest** Endothelial cells form the inner surface of blood vessels, acting like a non-stick coating. In addition to making substances that keep blood from sticking to the vessel wall, endothelial cells generate compounds that relax the vessel, and prevent it from thickening. Endothelial cells also form capillaries, the smallest vessels that provide oxygen and nutrients for all tissues.

A regenerating organ, or a bioengineered tissue, requires a system of capillaries and other microvessels. Thus, regenerative medicine could benefit from a knowledge of how to generate endothelial cells from pluripotent stem cells – cells that can "differentiate" to form almost any type of cell in the body.

Wong, Matrone et al. have now used a cell fusion model (named heterokaryon) to track the changes in gene expression that occur as a pluripotent stem cell differentiates to ultimately become an endothelial cell. In this model, mouse embryonic stem cells (ESCs) are fused to human endothelial cells. Over time the human endothelial cells drive gene expression in the ESCs toward that of endothelial cells.

Wong, Matrone et al. discovered changes in gene expression in many genes that have not previously been described as involved in the differentiation of endothelial cells. When one of these genes – named *Pou3f2* – was inactivated in ESCs, they could not be differentiated into endothelial cells. The absence of *Pou3f2* also drastically impaired how blood vessels developed in zebrafish embryos.

Thus the heterokaryon model can generate important information regarding the dynamic changes in gene expression that occur as a pluripotent cell differentiates to become an endothelial cell. This model may also be useful for discovering other genes that control the differentiation of other cell types.

## Results

### Interspecies heterokaryons as a discovery tool

To discover new genes involved in endothelial specification, we made heterokaryons consisting of human endothelial cells (hEC) and murine embryonic stem cells (mESC) (*Figure 1a–c*), which expressed cell surface markers and characteristics of both cell types. We hypothesized that the factors that are actively maintaining endothelial phenotype (transcription factors, epigenetic modifiers and non-coding RNA etc) would act on the pluripotent stem cell nuclei to induce expression of key determinants of endothelial lineage. We reasoned that we could use RNA seq to monitor global changes in the transcriptome of the pluripotent nucleus as it is reprogrammed in the heterokaryon toward an endothelial fate. In 95% of cases, the species-specific nucleotide differences between the mouse and human transcripts would permit us to differentiate between reads of murine versus human transcripts when the sequences were aligned to their respective genomes.

### Optimization and testing of the heterokaryon system

Reprogramming of the cell population is synchronized upon the addition of the fusagen. Since there is no nuclear fusion, chromosome rearrangement, or chromosome loss in the heterokaryons (*Bhutani et al., 2010*), we reasoned that this synchronization would permit us to study the temporal sequence of reprogramming to endothelial lineage using RNA seq. We optimized the cell fusion strategy using the fusagen HVJ (Sendai virus) envelope protein. By skewing the ratio of the input cells so that endothelial cells outnumbered pluripotent stem cells in the multinucleate heterokaryon, we forced reprogramming of the pluripotent stem cell nuclei toward an endothelial phenotype. To confirm that the system was working as anticipated, we assessed the expression of the mESC mRNA transcripts using murine-specific primers. In mESCs which were fused with human endothelial cells, we observed upregulation of murine endothelial genes including Kdr, Tie2, Cdh5 and Vwf (*Figure 1d–g*), and transcription factors involved in endothelial development such as Etv2, Ets1 and Tal1 (*Figure 1l–n*). Intriguingly, the expression of these genes seemed to mirror ontogeny, in that

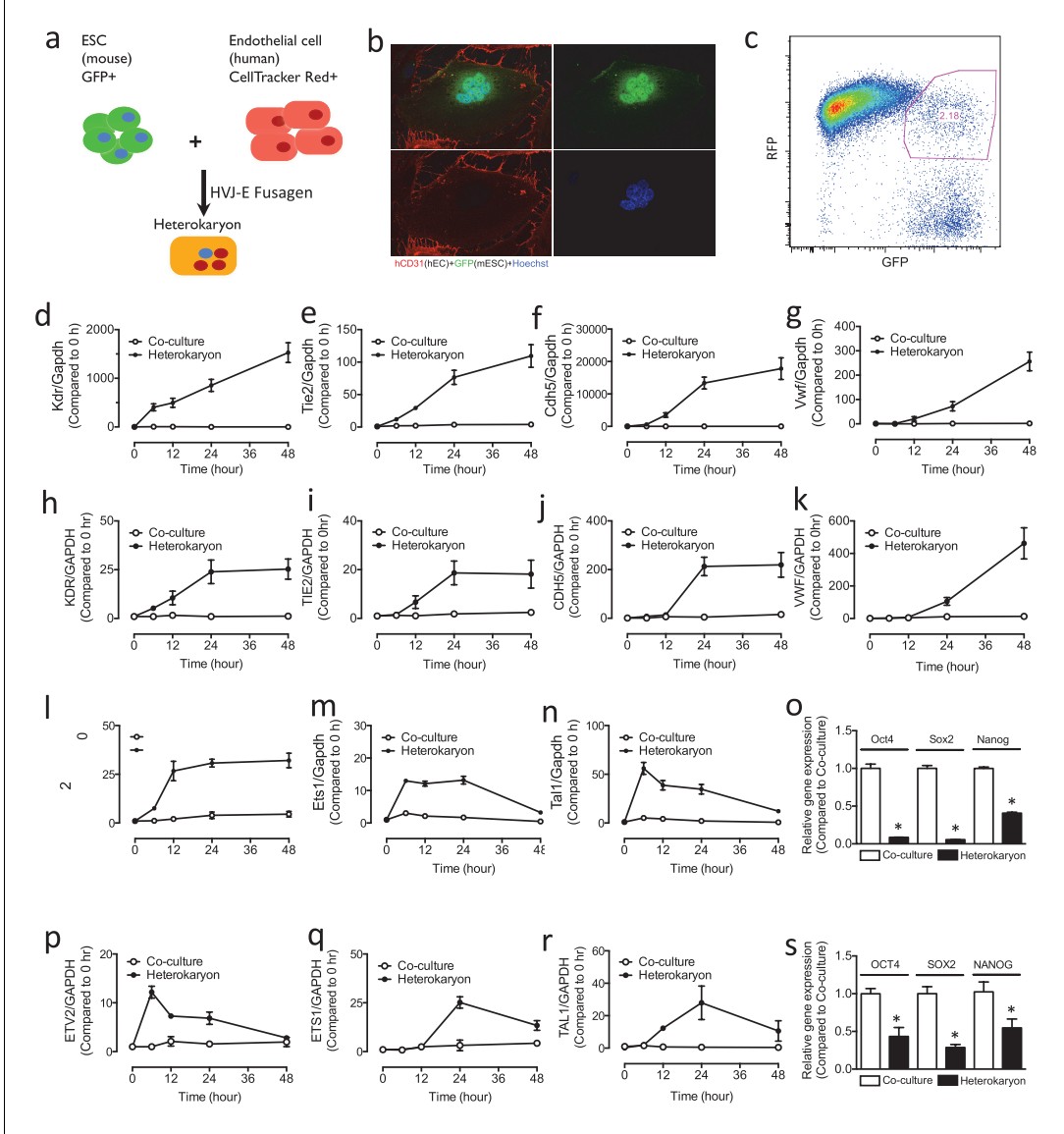

**Figure 1.** Heterokaryon recapitulates gene expression of endothelial ontogeny. (a) Scheme for heterokaryon generation. GFP-labeled murine ESCs (mESCs) were fused with Cell Tracker Red labeled human ECs (hECs) by HVJ-enveloped fusagen to induce multinucleated heterokaryons. (b) Representative image of non-dividing multinucleated heterokaryons labeled with CD31 (Red) and GFP (Green), Hoechst (Blue) dye were used to label nuclei. (c) Representative FACS plots for heterokaryons. (d–g) Up-regulation of murine EC genes including Kdr, Tie2, Cdh5 and Vwf in heterokaryons consisting of mESC and hEC compared to co-culture control. (h–k) Up-regulation of human EC genes including Kdr, Tie2, Cdh5 and Vwf in heterokaryons consisting of human iPSC (hiPSC) and murine EC (mEC) compared to Co-culture control. (l–n) Increased expression of transcription factors involved in endothelial development such as Etv2, Ets1 and Tal1 during cell fusion of mESC with hEC. (p–r) Increased expression of transcription factors involved in endothelial development such as Etv2, Ets1 and Tal1 during cell fusion of hiPSC with mEC. (o and s) Down-regulation of genes encoding pluripotent factors (Oct4, Sox2 and Nanog) in heterokaryons compared to Co-culture control. All data represented as mean ± S.E.M. (n = 3). p<0.05 vs Co-culture control.

genes involved early in mesoderm specification (e.g. Kdr) were expressed earlier in the heterokaryon, whereas genes that are more specific to hemato-endothelial lineage (e.g. Von Willebrand factor) were expressed later. In parallel to the upregulation of genes involved in endothelial development, we observed down-regulation of genes encoding pluripotent factors (Oct4, Sox2 and Nanog) (*Figure 1o*).

As a complementary approach, we generated heterokaryons consisting of murine endothelial cells (mEC) and human-induced pluripotent stem cells (hiPSC). We fused hiPSCs with mECs, and the expression of human mRNA transcripts was assessed using human specific primers. We observed in the pluripotent cells an upregulation of human genes including Kdr, Tie2, Cdh5 and Vwf (*Figure 1h–k*), and transcription factors involved in endothelial development such as Etv2, Ets1 and Tal1 (*Figure 1p–r*). Again, we observed a temporal sequence that mirrored ontogeny, with a parallel downregulation of genes encoding pluripotent factors (Oct4, Sox2 and Nanog) (*Figure 1s*). These results demonstrate rapid induction of endothelial genes in the heterokaryon which appears to recapitulate endothelial development.

## RNA seq for novel EC determinants

Having observed that the heterokaryon system seemed to mirror endothelial ontogeny, we applied RNA-seq to identify novel determinants of endothelial lineage (*Figure 2*). Heterokaryons were double-positive cells (GFP$^+$ for mESC and CellTracker Red$^+$ for hECs) and were harvested at 6–24 hr post-fusion. These time points were chosen based on our qPCR data, which showed rapid sequential activation of endothelial genes as early as 6 hr post-fusion. Total RNA (~2 ug) was isolated from heterokaryons or co-culture controls, and prepared for RNA-seq studies. We also did RNA-Seq for parental mESC; and for mESC that were exposed to an endothelial differentiation protocol for 4 or 8 days. We defined genes that are up- or down-regulated in each sample relative to the parental mESC. We were most interested in novel genes that encode transcriptional factors that are not known to be involved in mesodermal lineages or endothelial specification. We postulated that the heterokaryon system would be particularly useful to detect transcription factors that are transiently expressed during endothelial differentiation (*Figure 2a*). The RNAseq analysis of the heterokaryon samples (mESC/hEC) revealed a set of 1297 genes in the mESC that were upregulated as well as a set of 795 genes that were downregulated after fusion with hEC (*Figure 2b*). The upregulated genes included known endothelial related genes, for example, Kdr and Vwf, as well as hemangioblast markers, that is, angiotensin-converting enzyme (ACE, CD143). Conversely, murine pluripotency markers were down-regulated. We identified some novel transcriptional factors (such as Tbx1, Cebpd, Nrarp, Pou3f2, Maf, Tbx20, Tigd5, Pdn and Batf2), and targeted one of these (Pou3f2) for further study.

Unbiased hierarchical clustering analysis indicated that the transcriptome of the mESC in the heterokaryon is closer to that of mESCs differentiated toward EC lineage, than to the parental mESCs (*Figure 2c*). By contrast, the transcriptome of mESC co-cultured (but not fused) with hECs bears a closer relationship to the transcriptome of the parental mESCs. Global gene function enrichment analysis indicates that differentially up- or down-regulated genes in the mESC in heterokaryons are similar to those in mESC exposed to the endothelial differentiation protocol (*Figure 2d*). In particular, the upregulated genes in the heterokaryons tend to be involved in transcription regulation, RNA alternative splicing, DNA binding, embryonic organ development, differentiation and regulation of cell proliferation (*Figure 2e*), whereas downregulated genes are implicated in ATP and ribonucleotide binding (*Figure 2f*). These results suggest that the heterokaryon system may serve as an effective model for cell differentiation.

## Validation of Pou3f2 in differentiation to EC lineage

To validate our approach in discovery of novel determinants of cell lineage, we focused on Pou3f2, which was identified as a candidate EC transcription factor in the RNA seq studies. Accordingly, we examined the temporal sequence of gene expression of Pou3f2 in the heterokaryon system (*Figure 3a*). The expression of Pou3f2 was up-regulated at 6-hr post-fusion and its expression then declined over the ensuing 48 hr. In addition, we examined *Pou3f2* expression during the differentiation of murine ESC to endothelial cells using our standard differentiation protocol (*Figure 3b*). Intriguingly, the pattern of expression was similar, albeit with a slower time course. Specifically, the expression of Pou3f2 increased from Day 3 to Day 7 and then declined in the later phase (11 days) of the endothelial differentiation protocol. Notably, the expression of other mesodermal and EC genes followed a similar pattern, in that their regulation was accelerated when ESCs were placed into heterokaryons versus the standard endothelial differentiation protocol. This observation is consistent with the notion that current differentiation protocols could be considerably improved.

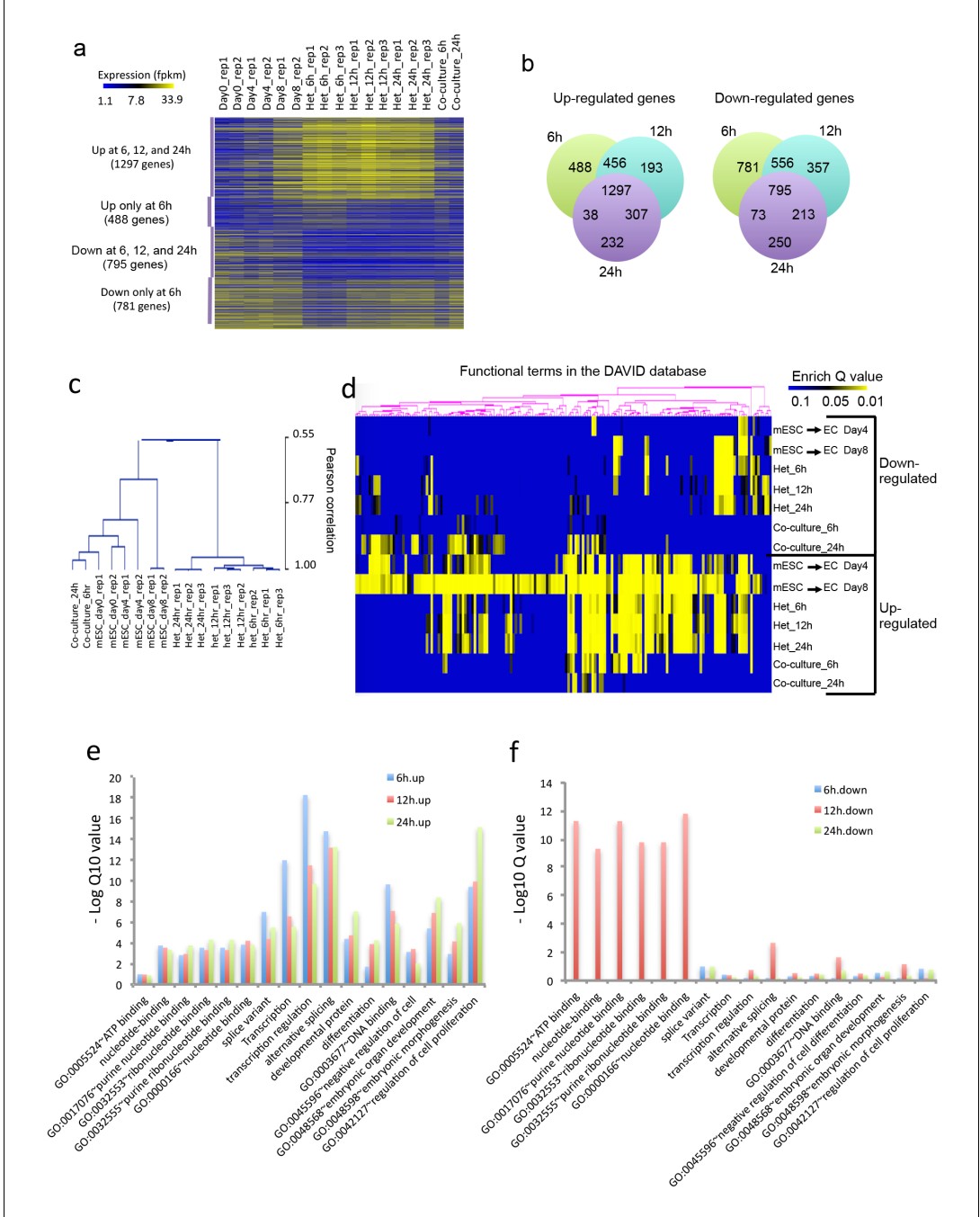

**Figure 2.** Heterokaryon bi-species RNA-seq reveals transcriptome reprogramming during differentiation of pluripotent stem cells into endothelial lineages. (a) Heat map to show expression level of genes that are up- or down-regulated either uniquely in the Heterokaryon (Het) at 6 hr after fusion or commonly in all het samples relative to the mESC samples. mESCs were exposed to a standard endothelial differentiation protocol for 4 or 8 days. Het_6 hr, Het_12 hr, and Het_24 hr indicate mouse stem cells (mESC) fused with human endothelial cells (hEC) for 6, 12 or 24 hr. Co-Culture_6 hr and Co-Culture_24 hr indicate mESC co-cultured but not fused with hEC. (b) Venn diagram to show overlap of upregulated (left) or downregulated (right) mouse genes in heterokaryon at 6, 12 and 24 hr after fusion relative to mESC. (c) Unbiased hierarchical clustering of all samples based on all genes that are differentially expressed in mESC samples relative to at least one of the other samples. (d) Heat map displaying upregulated and downregulated genes in heterokaryons and in differentiating mESC at different time points. Genes differentially expressed were clustered into groups for functional analysis and presented as a heat map based on their enrichment Q value.( e–f) Bar plot showing enrichment Q values of 17 functional terms in genes upregulated (left) or downregulated (right) in mESCs within heterokaryons relative to the mESCs. Upregulated or downregulated genes were defined based on EdgeR FDR cutoff 1e-5. Overlap p value in pie chart was calculated based on Fisher's Exact test. N = 3 for Het, n = 2 for mESCs, n = 1 for co-culture.

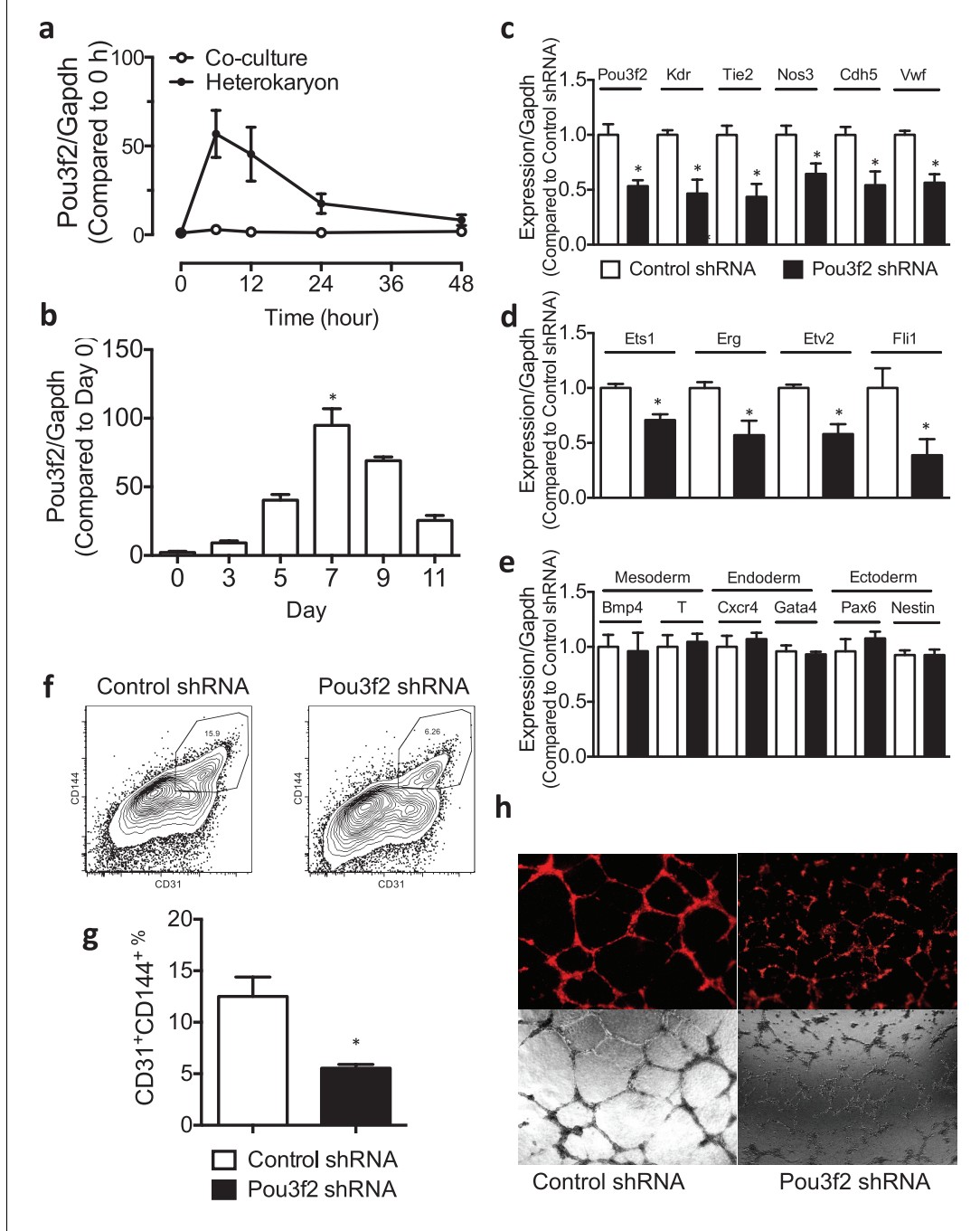

**Figure 3.** Role of Pou3f2 as a novel transcription factor in endothelial differentiation from murine embryonic stem cells. (**a**) Gene expression pattern of Pou3f2 in heterokaryons consisting of mESC and hEC compared to Co-culture control. (**b**) Validation of expression of Pou3f2 during differentiation of mESC into endothelial lineage. (**c**) Lentiviral mediated shRNA KD of Pou3f2 reduced the gene expressions of endothelial markers including Kdr, Tie2, Nos3, Cdh5 and Vwf at Day 8 following endothelial differentiation from mESC. (**d**) KD of Pou3f2 reduced the gene expressions of transcription factors involved in endothelial differentiation such as Ets1, Erg, Etv2 and Fli1 in mESC differentiated to endothelial lineage at Day 8. (**e**) No differences were found in the expressions of mesodermal (Bmp4, T), endodermal (Cxcr4 and Gata4) and ectodermal (Pax6 and Nestin) in Pou3f2 shRNA treated mESC following endothelial differentiation at Day 8. (**f** and **g**) Representative FACS plots and summarized diagram showing that Pou3f2 KD reduces the yield of mESC-derived CD31+ and CD144+ cells at Day 10 of endothelial differentiation protocol. (**h**) Representative images showing that Pou3f2 KD mESC-derived ECs manifest an impaired ability to form endothelial networks on matrigel. All data represented as mean ± S.E.M. (n = 3). p<0.05 vs control shRNA.

To further characterize the role of Pou3f2 in the differentiation of mESC into EC, we performed lentiviral shRNA knockdown (KD) of Pou3f2 in mESCs, and subjected the KD mESCs to the endothelial cell differentiation protocol. We found that KD of Pou3f2 reduced the expression of endothelial genes including Kdr, Tie2, Nos3, Cdh5 and Vwf at Day 8 of the differentiation protocol (*Figure 3c*). Similarly, the expression of endothelial transcription factors such as Ets1, Erg, Etv2 and Fli1 were also reduced in the Pou3f2 KD mESC (*Figure 3d*). The KD of Pou3f2 in mESC did not affect the expression of mesoderm- (Bmp4, T), endoderm- (Cxcr4, Gata4) or ectoderm- (Pax6 and Nestin) related genes (*Figure 3e*). Notably, the generation of mESC-derived CD31$^+$ and CD144$^+$ cells were reduced by over 50% in Pou3f2 *KD* group (*Figure 3g*). Furthermore, Pou3f2 *KD mESC*-derived endothelial cells manifested poor network formation on matrigel (*Figure 3h*). To summarize, Pou3f2 seems to be necessary for the full expression of genes known to be involved in endothelial development, and for the efficient generation of fully functioning endothelial cells. Amongst the factors released from endothelial cells in the heterokaryons that could control Pou3f2 expression there is Wnt and $\beta$-catenin. The Wnt/B-catenin signalling pathway is highly conserved and regulates vascular cell fate and development through Dll4/Notch signalling (*Corada et al., 2010*). The promoter for the Pou3f2 gene is a direct target for $\beta$-catenin/Lef1 (*Goodall et al., 2004a*). Endothelial cells in the heterokaryon might also contribute phosphatidylinositol 3-kinase to activate Pou3f2. The PI3K pathway mediates angiogenesis and the expression of growth factors in endothelial cells (*Jiang et al., 2000*) and also regulates Pou3f2 in melanoma cells (*Bonvin et al., 2012*).

## Importance of Pou3f2 in EC development

Pou3f2 promotes neurogenesis (*Jaegle et al., 2003*; *Castro et al., 2006*; *Dominguez et al., 2013*; *Sugitani et al., 2002*). Specifically, it activates the Notch ligand Delta1, synergistically with Mash1, to maintain a subset of neural progenitors in an undifferentiated state (*Castro et al., 2006*), whereas it suppresses the Notch effector Hes5 (*Dominguez et al., 2013*) that negatively regulate transcription of neurogenesis-promoting genes Neuregulins (*Imayoshi et al., 2008*). In human melanoma spheres and tumor xenograft, Pou3f2 is proposed to induce the Notch pathway (*Thurber et al., 2011*). In our studies, the importance of *Pou3f2* in endothelial development in vivo was assessed using the zebrafish model. The availability of transgenics expressing endothelial-specific fluorescent reporters, for example Tg(*fli1:EGFP*)$^{y1}$, combined with the transparency of the embryo, facilitate visualization of vascular development and blood flow in real time (*Baldessari and Mione, 2008*; *Ellertsdóttir et al., 2010*; *Holden et al., 2011*; *Kamei et al., 2010*).

In situ hybridization for *Pou3f2* showed that this transcription factor is expressed in most embryo structures, although the staining is particularly evident in the brain region (*Figure 4—figure supplement 1*). Because endothelial and hematopoietic cells share a common progenitor, we assessed the expression of *Pou3f2* in both endothelium and hematopoietic cells. To do so, we purified GFP+ cells from *tg(fli1:EGFP)*$^{y1}$ or *tg(cmyb:GFP)* embryos. We found expression of *Pou3f2* in both type of cells, although more pronounced in the GFP+ cells from *tg(fli1:EGFP)*$^{y1}$ embryos (*Figure 4—figure supplement 2*). Injection of morpholino targeting *Pou3f2* (the relevant zebrafish analogue) resulted in 100% embryo death, as embryo could not reach 50% epiboly stage, at 5.3 hr post-fertilization. Both Pou3f2-targeted morpholinos caused the same effects. Therefore, we used light-activated cyclic caged morpholinos against *Pou3f2* to study the role of this gene in vascular development (*Figure 4a–d*). When the Pou3f2-targeted morpholino was activated by UV light at six hpf the embryo survival was around 80%. We observed a general phenotype characterized by curved body and curly tail, and specific dysmorphogenesis of the vascular system in most of the *tg(fli1:EGFP)*$^{y1}$ embryos (*Figure 4a*). A reduced number of intersegmental vessels (ISVs) was observed (*Figure 4b–c*), which could be rescued with *Pou3f2* mRNA. In situ hybridization showed that *Fli1* and *Kdr* were downregulated following *Pou3f2* KD, particularly evident in the intersegmental vessels, confirming the role for *Pou3f2* in endothelial development (*Figure 4d*). These results were confirmed by real-time PCR (*Figure 4—figure supplement 3*), where *Fli1* and *Kdr* were significantly reduced in *Pou3f2* knockdown embryos compared to control. Effective knockdown by the morpholino in zebrafish embryos was documented by Western blotting using a *Pou3f2*-specific antibody (*Figure 4e*). In addition, there was a global reduction of endothelial cells in *Pou3f2* KD animals. Specifically, the percentage of GFP$^+$ cells isolated from the *tg(fli1:EGFP)*$^{y1}$ zebrafish embryos at 24 hr post-fertilization was 6.7% of total cells in control compared to 3.7% following *Pou3f2* KD (*Figure 4f–g*). Injection of Pou3f2-targeted caged morpholino in tg(*cmyb:GFP*) did not significantly reduce the percentage of

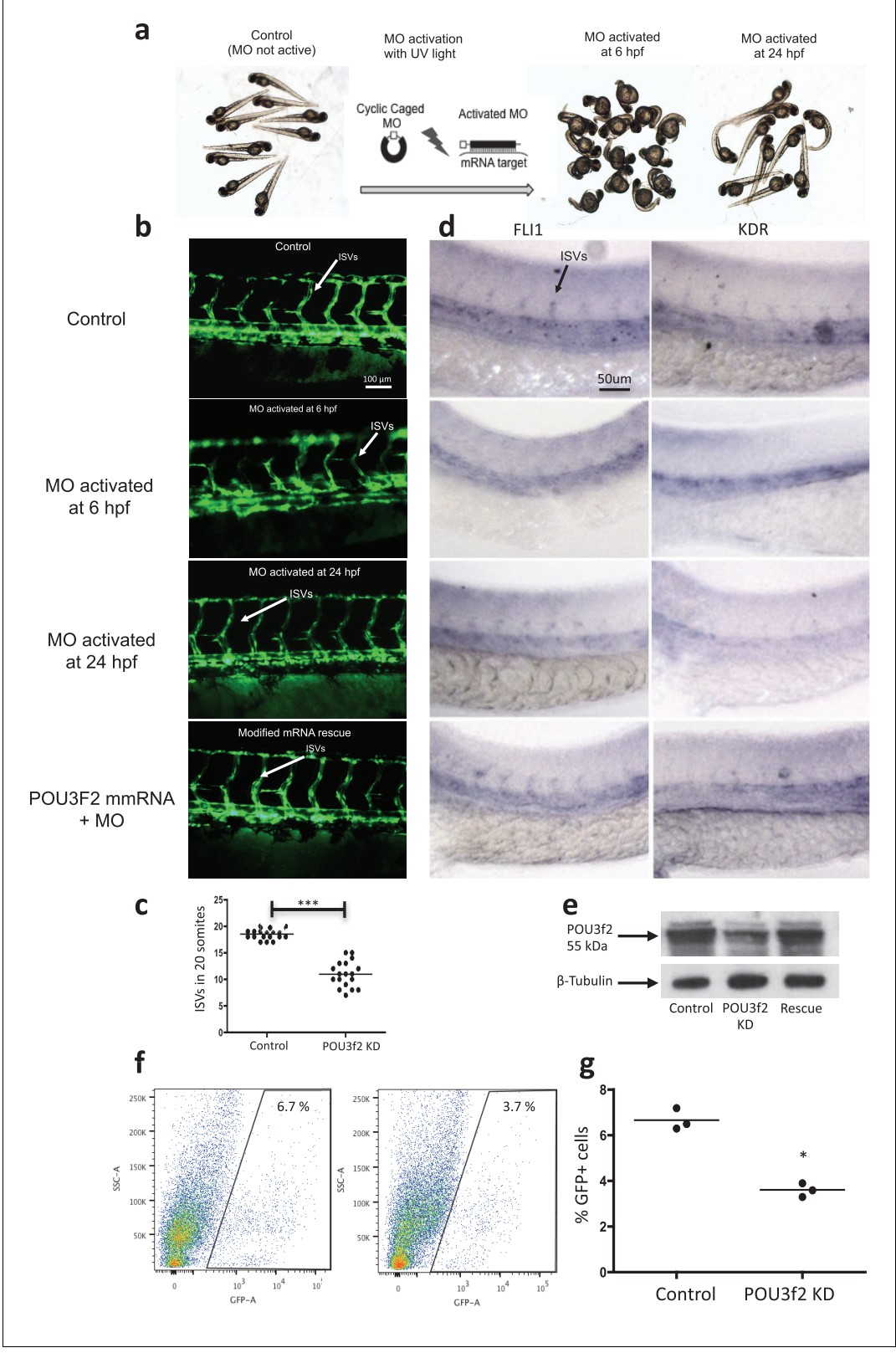

**Figure 4.** Pou3f2 knockdown in the *tg(fli1:EGFP)*[y1] zebrafish embryo. (a) Bright-field images of embryos injected with caged morpholino against *Pou3f2* translation start site in the absence of photoactivation (control), or with photoactivation with UV light at 6 or 24 hpf. (b) Fluorescence images of embryos at 48 hpf. Experimental groups were injected with caged morpholino against *Pou3f2* in the absence of photoactivation (control), or with

*Figure 4 continued on next page*

*Figure 4 continued*

photoactivation with UV light at 6 or 24 hpf, or with photoactivation at 6 hr in the presence of rescue mRNA encoding *Pou3f2*. (c) Quantitation of the number of intersegmental vessels in 20 somites in embryos at 48 hpf. (d) In situ hybridization with antisense RNA probes specific for *Kdr* and *Fli1* in whole zebrafish embryos 28 hpf. (e) Western blotting showing the reduction level of *Pou3f2* following morpholino injection and rescue by mRNA encoding Pou3f2. $\beta$-Tubulin was used as loading control. ISV – Intersegmental Vessels; hpf – hour post fertilization. (f and g). Representative FACS plot and scatter plot showing a significant reduction of GFP$^+$ cells in *Pou3f2* KD embryos. GFP$^+$ cells were sorted following isolation by enzymatic digestion from *tg(fli1:EGFP)$^{y1}$* zebrafish embryos at 24 hpf. All data represented as mean ± S.E.M. N = 3. Student t-test, *p=0.01; ***p=0.001.

The following source data and figure supplements are available for figure 4:

**Source data 1.** Intersegmental vessel analysis in zebrafish embryos following Pou3f2 knockdown.
**Figure supplement 1.** In-situ hybridization for Pou3f2 in zebrafish embryos.
**Figure supplement 2.** Pou3f2 gene expression in endothelial and hematopoietic lineages.
**Figure supplement 2—source data 1.** Real time PCR analysis of Pou3f2 in zebrafish embryo Fli1+ and Cmyb+ cells.
**Figure supplement 3.** Gene expression of endothelial markers following Pou3f2 knockdown.

GFP+ cells, suggesting that *Pou3f2* is more critical for endothelial rather than hematopoietic development. Notably, caged MO activation at 24 hpf or later did not produce a significant phenotype, indicating that *Pou3f2* is necessary early in endothelial development, but is not required for maintenance of the endothelial phenotype. This observation is consistent with the lack of expression of Pou3f2 in mature endothelial cells. Our studies suggest that Pou3f2 could act upstream of Notch during endothelial, as well as neuronal development. In this regard, Pou3f2 could have a role in the human vascular diseases caused by mutations in Notch family members, such as Alagille syndrome and cerebral autosomal dominant arteriopathy with subcortical infarcts and leukoencephalopathy (CADASIL) (*Loomes et al., 1999*; *Joutel et al., 1997*). As Notch signaling is also required for arterio-venous differentiation during vascular development (*Lawson et al., 2001*), this suggests that Pou3f2 could be also upstream of the eph-ephrin system and play a role in arterio-venous specification. Pou3f2 gene expression seems to be controlled by MAPK (*Goodall et al., 2004b*) and Wnt/$\beta$-catenin (*Goodall et al., 2004a*) signaling, two key pathways linked to cell proliferation. Pou3f2 function can be also modulated by posttranslational modifications including sumoylation, ubiquitinylation, glycosylation and in particular phosphorylation (*Kasibhatla et al., 1999*; *Augustijn et al., 2002*; *Diamond et al., 1999*; *Nieto et al., 2007*).

## Role of Pou3f2 in human EC differentiation

To determine the role of Pou3f2 during endothelial differentiation from human iPSC, we examined the expression of Pou3f2 in the heterokaryon system consisting of hiPSCs and murine endothelial cells (*Figure 5a*). In addition, we examined the expression of Pou3f2 in the hiPSC during our endothelial differentiation protocol (*Figure 5b*). The expression patterns for Pou3f2 were similar to that which we observed in mESC (*Figure 3a and b*). Notably, we saw the same accelerated regulation of Pou3f2 in the hiPSCs in the heterokaryon by comparison to hiPSCs exposed to the endothelial differentiation protocol.

Next, we assessed the effects of shRNA-mediated KD of Pou3f2 on differentiation of endothelial cells from human iPSC. Pou3f2 *shRNA* significantly reduced the expression of Pou3f2 during the endothelial differentiation protocol (*Figure 5c,d*). Furthermore, Pou3f2 *KD* significantly inhibited the expression of endothelial-related genes during the course of the differentiation protocol, including Kdr, Tie2, CDdh5 Pecam1, Nos3 and Vwf (*Figure 5e–j*). Notably, Pou3f2 KD also reduced iPSC-EC generation by ~50% (*Figure 6a–c*). In addition, protein expression of CD31, CD144 and Vwf (*Figure 6d*) and formation of tubular networks on matrigel (*Figure 6e*) were reduced in ECs derived

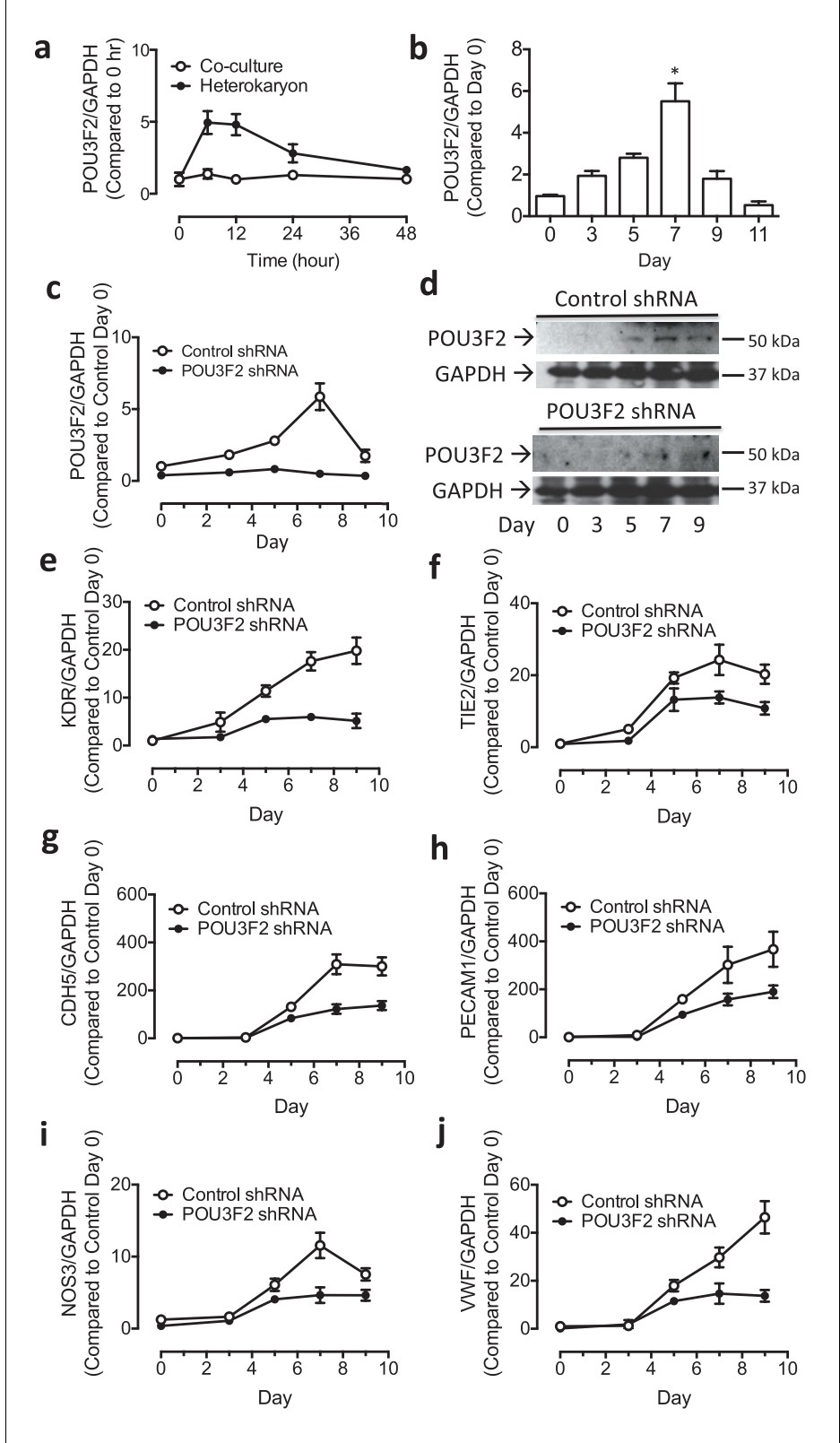

**Figure 5.** Role of Pou3f2 as a novel transcription factor in endothelial differentiation from human-induced pluripotent stem cells. (a) Gene expression pattern of Pou3f2 in heterokaryons consisting of hiPSC and mEC compared to co-culture control. (b) Validation of expression of Pou3f2 during differentiation of hiPSC into endothelial lineage. (c) Expression of Pou3f2 in lentiviral mediated shRNA KD of Pou3f2 in hiPSC following
*Figure 5 continued on next page*

*Figure 5 continued*

differentiation into endothelial phenotype compared to Control shRNA group. (**d**) Representative images of Western blots showing the KD effects of Pou3f2 in hiPSC during endothelial differentiation, the same results were obtained at least three times. (**e–j**) Pou3f2 KD reduced the gene expression of endothelial markers including Kdr, Tie2, Cdh5, Pecam1, Nos3 and Vwf following endothelial differentiation of hiPSC. All data represented as mean ± S.E.M. (n = 3). p<0.05 vs co-culture control or control shRNA group.

from Pou3f2 KD iPSCs. Furthermore, Pou3f2 KD iPSC-ECs exhibited impairment of other endothelial functions including nitric oxide generation (*Figure 6f*), and uptake of acetylated LDL (*Figure 6g*). These functional impairments were associated with reduced expression of EC markers including Kdr, Tie2, Nos3, CD31, Cdh5 and Vwf (*Figure 6h*). We have previously shown that endothelial cells derived from iPSCs (hiPSC-ECs) expressed various markers associated with arterial, venous and lymphatic ECs and thereby represent a heterogeneous population of ECs (*Rufaihah et al., 2011*). We found that ECs generated from Pou3f2 KD iPSCs resemble those generated from wild-type iPSCs with respect to venous (Ephb4 and Coup-TFII) and lymphatic markers (Pdpn and Lyve1) but have a significant reduction of arterial markers (Notch4, Efnb2 and Hey2) compared to scrambled control (*Figure 6i*).

## Pou3f2 interacts with known endothelial promoters

To determine if Pou3f2 binds to the promoters of endothelial-related transcription factors, we performed ChIP-PCR. We observed that during differentiation of iPSCs to ECs, the binding of Pou3f2 to the promoters of endothelial related transcription factors including Ets1, Lmo2, Hey1 and Hey2 (*Figure 7a–d*) was inhibited in Pou3f2 KD cells, in association with reduced gene expression of these factors (*Figure 7f–i*). Finally, we observed that the generation of CD31$^+$CD144$^+$ cells was reduced in the Pou3f2 KD hiPSCs and could be rescued with modified mRNA encoding Pou3f2 (*Figure 7j*).

## Discussion

Our current understanding of the genetic and epigenetic processes governing endothelial development and differentiation is limited. We lack comprehensive knowledge regarding all endothelial lineage factors and have sparse information regarding the magnitude and temporal sequence of their expression. In this paper, we find that the bi-species heterokaryons combined with RNAseq can provide new insights into determinants of endothelial lineage. Our work suggests that transcription factors and epigenetic machinery which actively maintain endothelial phenotype can also act on the pluripotent cell nucleus to recapitulate ontogeny. This system is likely to generate useful insights to improve the yield and fidelity of reprogramming to endothelial phenotype. A tangible and immediate outcome of this line of inquiry will be a more complete knowledge of the hierarchy of genes regulating differentiation to the EC lineage. Insights into these processes will be of general interest to investigators of vascular differentiation and development and may lead to new therapeutic targets for endothelial regeneration and the treatment of vascular diseases.

Finally, and perhaps most importantly, this model system should be amenable to discovery of novel determinants of other cell lineages. We believe that our studies provide proof-of-concept for using bi-species heterokaryon technology as a tool to elucidate novel genes regulating differentiation to any somatic cell. Our work opens a new vista of exploration for the broader community of scientists working in tissue regeneration, development, differentiation and the therapeutic applications of these insights.

## Materials and methods

### Cell culture

Human-induced progenitor stem cells (Takara Bio USA, Mountain View, CA) line authentication was achieved by genetic profiling using polymorphic short tandem repeat (STR profiling) loci. Our cell cultures were tested weekly for mycoplasma by real-time PCR approach and were mycoplasma-free.

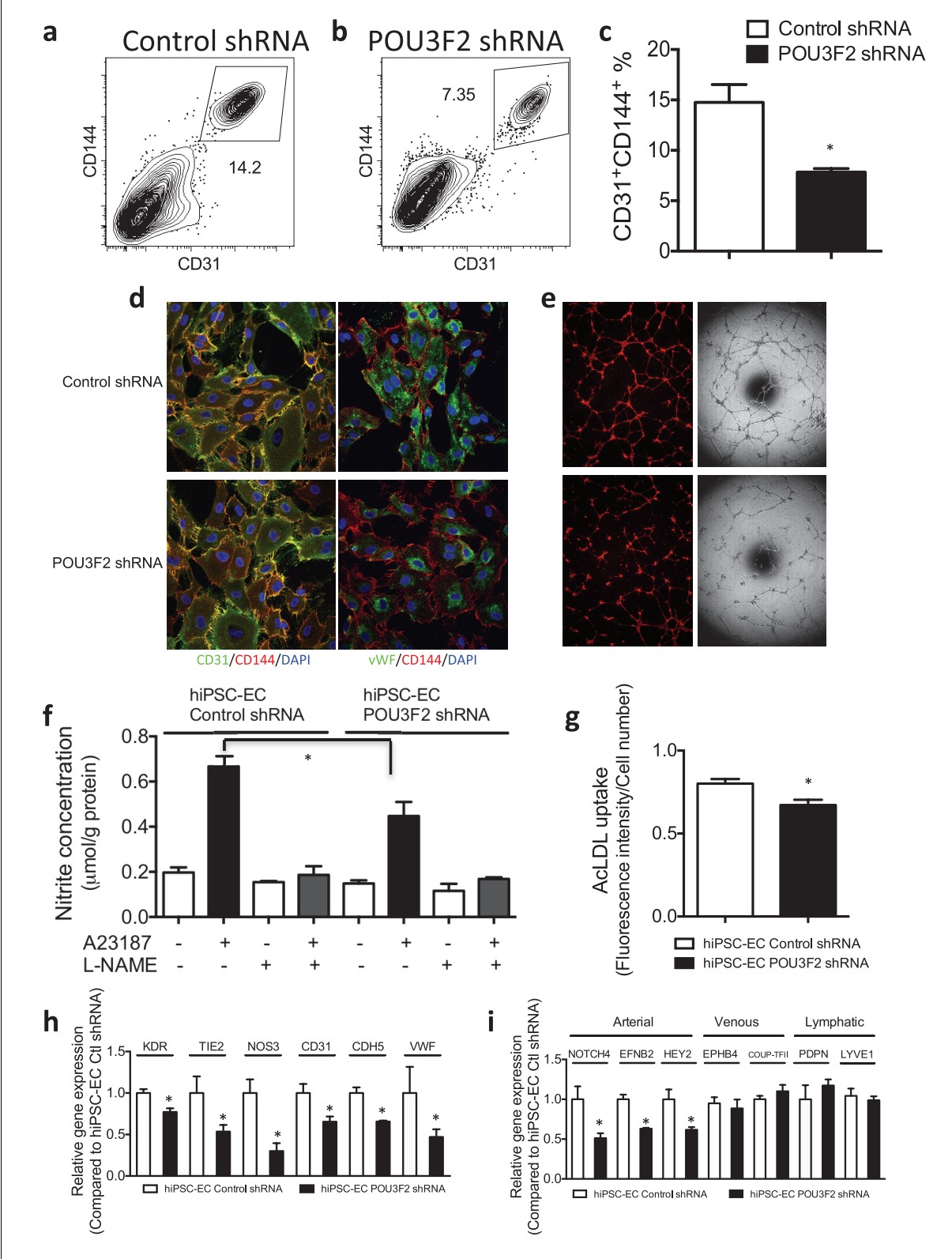

**Figure 6.** Functional assays further reveal the importance of Pou3f2 during differentiation of human iPSC into endothelial phenotype. (a–c) Representative FACS plots and summarized diagram showing Pou3f2 KD reduced iPSC-EC generation compared to scrambled control. (d) Representative immunofluorescence images revealed lower expression of CD31, CD144 and Vwf in Pou3f2 KD iPSC-ECs. (e) The iPSC-ECs generated from Pou3f2 KD cells manifested poor formation of networks of tubular structures on matrigel. (f) The ability of Pou3f2 KD iPSC-ECs to produce nitric

*Figure 6 continued on next page*

*Figure 6 continued*

oxide in response to calcium ionophore A23187 was significantly reduced compared to scrambled control iPSC-ECs. (**g**) Reduced capacity in taking up AcLDL in Pou3f2 KD iPSC-ECs compared to scrambled control iPSC-ECs. (**h**) Reduced gene expression of endothelial markers including Kdr, Tie2, Nos3, CD31, Cdh5 and Vwf in Pou3f2 KD human iPSC-ECs. (**i**) Arterial markers (Notch4, Efnb2, Hey2) but not venous (Ephb4 and Coup-TFII) nor lymphatic markers (Pdpn and Lyve1) were affected in Pou3f2 KD iPSC-ECs compared to scrambled control iPSC-ECs. All data represented as mean ± S.E.M. (n = 3). p<0.05 vs Control shRNA group.

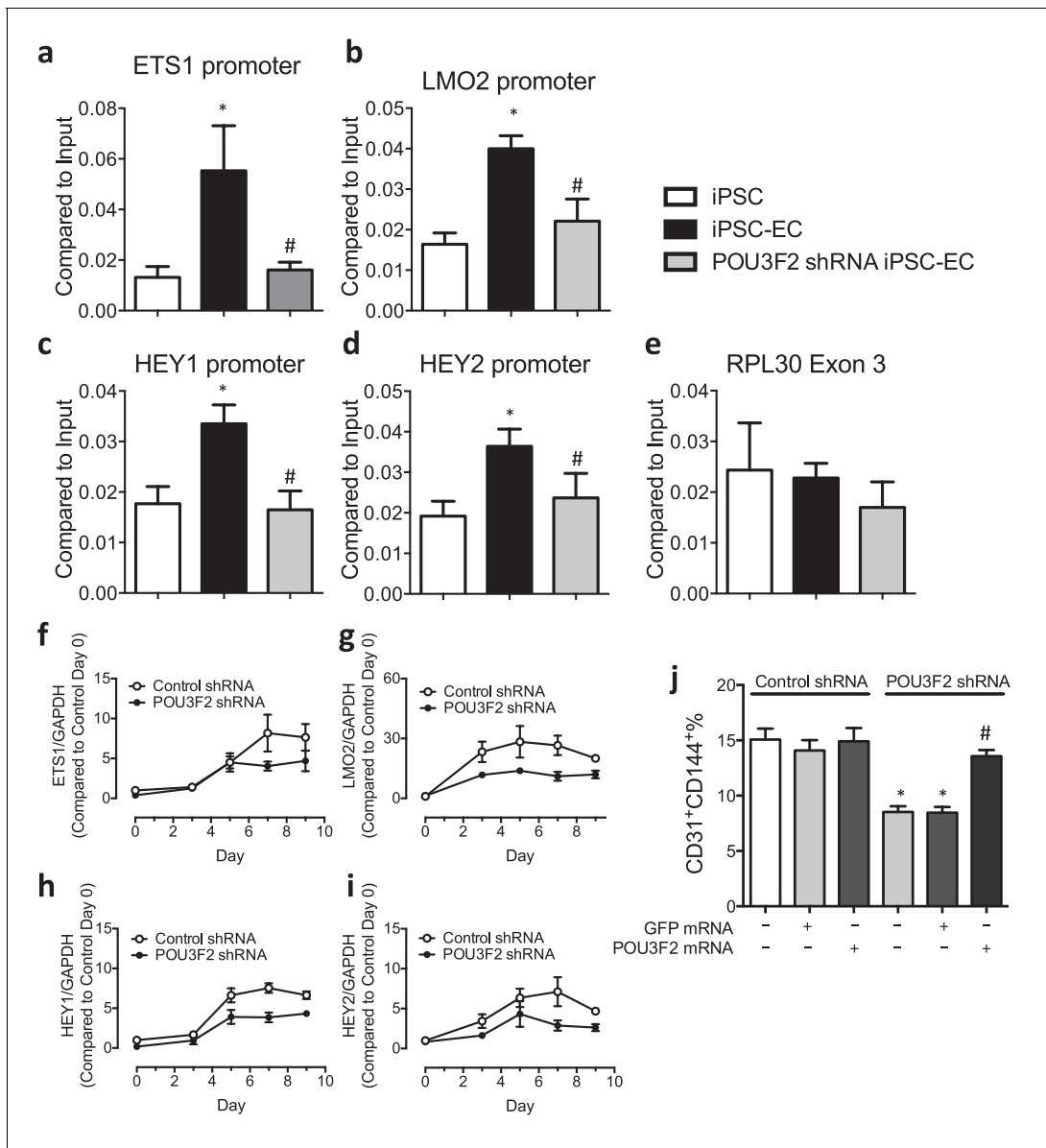

**Figure 7.** Chromatin immunoprecipitation (ChIP) and qPCR analysis reveal Pou3f2 binds to the promoters of endothelial-related transcription factors. (**a–d**) Binding of Pou3f2 to the promoters of endothelial-related transcription factors including Ets1, Lmo2, Hey1 and Hey2 was significantly inhibited in Pou3f2 KD cells compared to scrambled control at Day 8 of endothelial differentiation protocol, without affecting the control promoter RPL30 Exon 3 (**e**). (**f–i**) Downregulation of gene expression of Ets1, Lmo2, Hey1 and Hey2 in Pou3f2 KD cells during differentiation into endothelial lineage. (**j**) Rescue experiments with modified mRNA encoding Pou3f2 improved CD31⁺CD144⁺ cell generation from Pou3f2 KD cells. All data represented as mean ± S.E.M. (n = 3). p<0.05 vs co-culture control or control shRNA group.

HiPSC lines were generated using retroviral factors encoding Oct4, Sox2, Cmyc and Klf4 in adult dermal fibroblasts. The hiPSCs were characterized for their pluripotency using PCR and IHC for known pluripotency markers, and were maintained in mTeSR1 (Stem Cell Technology, Vancouver, Canada). Murine ESCs (D3, ATCC, Manassus, VA) of the SV129 strain were cultured on gelatin-coated dish and maintained in ESGRO media plus GSK3$\beta$ inhibitors. Human microvascular endothelial cells (HMVECs) (obtained from Lonza, Walkersville, MD) and murine endothelial cells (obtained from Applied Stemcell, Menlo Park, CA) were cultured in EC growth medium EGM-2 MV (cc-3162). Cells were used for all experiments at passage 6–8.

## Heterokaryon formation and isolation

One day before cell fusion, $4 \times 10^5$ endothelial cells were seeded on one well of a 6-well dish. The endothelial cells were confluent on the day of cell fusion. On the same day, 1 hr prior to cell fusion, the endothelial cell medium was replaced with fresh EGM-2 MV medium supplemented with 1 µM Cell Tracker Red, then cells were incubated at 37°C in darkness for 30 min. Human iPSCs or murine ESCs labeled by transduction with retroviruses encoding GFP were rinsed with PBS followed by accutase treatment at 37°C for 5 min to dissociate the pluripotent stem cells into single-cell suspension. The cells were then collected in conical tubes after neutralization by MEF media containing 10% FBS. The cells were then counted with hemocytometer and $2 \times 10^5$ pluripotent stem cells were taken. The cells were then centrifuged at 200 x g for 5 min at 4°C, the supernatant was removed, and the cell pellet was resuspended in 25 µL ice-cold cell fusion buffer with 2.5 µL ice-cold HVJ-envelope fusagen. The reaction mixture was placed on ice for 5 min with regular agitation in 2.5 min apart. After 5 min, the cells were centrifuged again and the supernatant was discarded and 2 ml cell fusion buffer was added. The pluripotent cells were then plated onto the endothelial cells. The six-well dish was then centrifuged at 200 g for 5 min at 4°C. After centrifugation the dish was placed into a 37°C incubator to induce cell fusion. Twenty minutes later, the medium was removed and EGM-2 MV medium was added. For the Co-culture Control, the described procedure was the same except HVJ-enveloped fusagen was not added. The heterokaryons (double-positive cells) can be efficiently sorted by FACS. Heterokaryons (GFP$^+$ and CellTracker Red$^+$) were harvested by FACS at 6, 12, 24, 48 and 72 hr post-fusion.

## Preparation of RNA-seq libraries and sequencing

The species-specific nucleotide differences between the mouse and human transcripts enable us to differentiate between reads of transcripts from the murine ESC versus those from the human EC when the sequences are aligned to their respective genomes. Heterokaryons (GFP$^+$ and CellTracker Red$^+$) were harvested by FACS at 6, 12 and 24 hr post-fusion and prepared for analysis by RNA-seq. Total RNA from heterokaryons were isolated. Human and mouse mRNA transcripts were isolated from the total RNA samples using polyA-based enrichment using oligo-dT magnetic beads. The majority of contaminating ribosomal RNA was eliminated by this approach. The resulting mRNA was fragmented, reverse transcribed to cDNA, ligated to adapters, and subject to brief PCR amplification in preparation of the Illumina library. The integrity and quality of RNA and complementary DNA were monitored using an Agilent Bioanalyzer 2100. The samples were sequenced using pair-end 100 base-pair reads. For the estimation of gene expression and data analysis, any remaining ribosomal reads were discarded, and the resulting murine and human transcripts were mapped to their respective genomes. Reads that map to both transcriptomes would be discarded and the RPKMs adjusted accordingly (discarded reads represent only 5% of the total reads; furthermore, virtually all genes have at least one unique read that is different between species, so that no gene is completely discarded).

## RNA-seq read mapping and gene expression

RNA-Seq reads were aligned to the mouse genome version mm9 using TopHat version 2.1.0. We use the full set of knownGene downloaded from the UCSC Genome browser (http://genome.ucsc.edu/cgi-bin/hgTables) as reference genes. RNA-Seq read counts for each gene in each sample was calculated using Cuffdiff function in Cufflinks version 2.2.1. The Cuffdiff also calculates fragment per kilobase per million reads (FPKM) for each gene. We further subject the reads counts to EdgeR version 3.12.0 for differential expression analysis, and define differential genes based on false discovery

rate (FDR) cutoff 1e-5. We subject interesting gene groups to the DAVID website (https://david. ncifcrf.gov) for functional enrichment analysis. Enriched functional terms were defined based on Benjamini adjusted p value cut-off 0.05. Hierarchical clustering of gene expression heatmap was conducted using MEV based on Pearson correlation distance metric and the average linkage method.

## RNA extraction and quantitative PCR

Using RNeasy Mini Kit (Qiagen, Chatsworth, CA), total RNA was extracted. The Quantitect reverse transcription kit (Qiagen) was used to generate cDNA and SYBR Green PCR kit (Invitrogen, Carlsbad, CA) was used for real-time qPCR with the QuantStudio 12 k Flex system (Applied Biosystems, Foster City, CA) following the manufacturer's instructions. Genes were analyzed with the data normalized to Gapdh and expressed as relative fold changes using the ΔCt method of analysis.

## Endothelial differentiation protocol

Murine ESC-EC Differentiation: Endothelial differentiation of ESCs was carried out using the suspension culture approach with modifications. To initiate differentiation, ESCs were cultured in ultralow nonadhesive dishes to form embryoid body aggregates in a differentiation medium that consisted of α-Minimum Eagle's Medium, 10% FBS, 1% penicillin/streptomycin, and 0.05 mmol/L $\beta$-mercaptoethanol (Sigma, St Louis, MO). After 4 days of suspension culture, the embryoid bodies were reattached onto 0.2% gelatin-coated dishes and cultured in differentiation medium. After 3 weeks of differentiation, the cells were purified by fluorescence-activated cell sorting (FACS) using anti-mouse vascular endothelial cadherin (VE-cadherin) antibody (Ab) (BD Biosciences, Bedford, CA).

Human iPSC-EC differentiation: Confluent cultures of hiPSCs were incubated with 1 mg/ml type IV collagenase for 10 min and transferred to ultra low attachment dishes containing differentiation media for 4 days to form embryoid bodies (EBs). The differentiation media used consisted of α-Minimum Eagle's Medium, 20% fetal bovine serum, L-glutamine, $\beta$-mercaptoethanol (0.05 mmol/L) and 1% non-essential amino acids supplemented with bone morphogenetic protein-4 (BMP-4, 50 ng/ml, Peprotech) and vascular endothelial growth factor (VEGF-A, 50 ng/ml, Peprotech). The four-day EBs were reattached to gelatin-coated dishes in the presence of VEGF-A for another 10 days before purification.

## Fluorescence-activated cell sorting (FACS)

ECs derived from pluripotent stem cells were purified using FACS. Cells were dissociated into single cells with Accutase (Invitrogen) for 5 min at 37°C, washed with 1x PBS containing 5% BSA and passed through a 70-µm cell strainer. Cells were then incubated with either Alexa Fluor 488-conjugated CD31 antibody (BD Bioscience, San jose, CA) or PE-conjugated CD144 antibody (BD Bioscience) for 30 min. Isotype-matched antibody served as negative control. The purified ESC- or iPSC-ECs were expanded in EGM-2 media.

## Immunofluorescent imaging

Human iPSC-ECs were fixed with 4% paraformaldehyde, permeabilized with 0.1% Triton X-100, blocked with 1% normal goat serum and stained for anti-human CD31 (R and D Systems), anti-human CD144 (R &D Systems, Minneapolis, MN), anti-human von Willebrand factor (vWF, Abcam, Cambridge, UK) overnight at 4°C. After washes with PBS, the cells were treated with Alexa Fluor-488 or -594 secondary antibodies. Cell nuclei were stained with Hoechst 33342 (Sigma). Images were acquired on a confocal microscope (FV1000-IX81, Olympus, Tokyo, Japan).

## Western blotting

Cells were homogenized with ice-cold RIPA lysis buffer containing 1 µg/mL leupeptin, 5 µg/mL aprotonin, 100 µg/mL PMSF, 1 mmol/L sodium orthovanadate, 1 mmol/L EDTA, 1 mmol/L EGTA, 1 mmol/L sodium fluoride and 2 µg/mL $\beta$-glycerolphosphate. The protein concentration was determined by Bradford method and aliquots of 20 µg of the total proteins were separated on 10% SDS-poly-acrylamide gel. Proteins were then transferred to immobilon-P polyvinylidene difluoride (PVDF) membrane (Millipore, Billerica, MA). Membranes were blocked with 5% non-fat milk in TBS-T and subsequently exposed to *Pou3f2* primary antibody (Genetex, Irvine, CA) followed by HRP-conjugated secondary antibody and developed by chemiluminescence.

## Functional assays

Uptake of Ac-LDL: was evaluated by incubating cells with ac-LDL-594 at 1:200 dilution for 5 hr before washing the cells with PBS and then measuring the mean fluorescence of the cells. Endothelial network formation, the ability of cells to form tube-like structures, was assessed *in vitro* by seeding $1.2 \times 10^5$ cells in wells coated with matrigel in the presence of EGM-2 media containing 50 ng/ml VEGF and incubated for 24 hr.

## Nitric oxide production

The ability of the cells to produce NO was assessed by measuring the concentration of NO in the culture medium using the NO detection kit (Molecular Probe, Carlsbad, CA) according to the manufacturer's instructions. The amount of nitrate was determined by converting it to nitrite, followed by the colorimetric determination of the total concentration of nitrite as a colored azo dye product of the Griess reaction that absorbed visible light at 540 nm using a microplate reader.

## Chromatin immunoprecipitation and ChIP-qPCR

hiPSCs were differentiated towards EC lineage and collected at Day 8. Samples were prepared by SimpleChIP enzymatic chromatin IP kit (Cell Signaling Technology). Chromatin immunoprecipitation was performed using human Pou3f2 antibody (Genetex), rabbit IgG (CST), histone H3 antibody (CST). DNA was purified using Nucleospin PCR clean-up kit (Macherey-Nagel, Bethlehem, PA) and used for quantitative PCR with primers against regions predicted within the promoter of Ets1, Lmo2, Hey1 and Hey2. Recovery of genomic DNA as the percentage input was calculated as the ratio of copy numbers in the immunoprecipitate to the input control.

## Zebrafish aquaculture and husbandry

Adult zebrafish (wild-type Wik and *tg(fli1:EGFP)$^{y1}$* strains) were acquired from the Zebrafish International Resource Center and raised according to standard procedures and kept at 28°C under a 14/10 hr light/dark cycle and fed with dry meal (Gemma Micro, Westbrook, ME) twice per day. Embryos used in these studies were obtained by natural matings and cultured in E3 embryo medium at 28.5°C. Animals were housed and all experiments were carried out in accordance with the recommendations of the Institutional Animal Care and Use Committee. All surgery procedures were performed under anesthesia with Tricaine 0.02 mg/ml.

## Morpholinos and caged morpholinos injections

*Pou3f2* KD in zebrafish was achieved using two different antisense morpholinos (Gene Tools, Oregon) targeting the *Pou3f2* mRNA AUG translational start site with sequence: (Mo1) 5'-ATGATTGGATGC TGTAGTCGCCATG-3', and (Mo2) 5'-CGGACTGATCGCTCCTATTAAAGGA-3'. As one control we used a 5-base pair mismatch MO: sequence 5'-ATcATTcGATcCTGTAcTCcCCATG-3'. To decipher the roles of *Pou3f2* transcription factor in specific stages of endothelial development, we used *Pou3f2*-targeted caged morpholino (cMOs) (*Shestopalov et al., 2007*). This chemically modified morpholino allowed temporal gene silencing by using targeted UV illumination. An optimized dose of 0.5 ng/eggs (0.5 nL bolus) of *Pou3f2* targeted morpholino was injected in each embryo at 1–2 cell stage, just below the cell mass.

## cMOs photoactivation

To photoactivate the cMOs, injected zebrafish embryos were arrayed in an agarose microinjection template (560 µm x 960 µm wells), with the animal pole facing the light source. Then, the mercury lamp light was focused onto individual embryos for 10 s, using a Leica DM4500B epifluorescence microscope equipped with an A4 filtercube (Ex: 360 nm, 40 nm bandpass) and a 20 x/0.5 NA water-immersion objective. Individual embryos were irradiated at 6 or 24 hr post-fertilization.

## Morpholino phenotype rescue by modified mRNA

As control, we also performed rescue experiments by co-injecting *Pou3f2*-targeted MO together with *Pou3f2* modified mRNA into one-cell-stage embryos. The *Pou3f2* modified mRNA version used, produced from the RNA Core available in our Institute, was modified at the 5' untranslated region

so that it was not recognized by the morpholino. An optimized dose of 300 pg was co-injected with the morpholino in rescue experiments.

## Zebrafish imaging

The embryos were manually dechorionated at 24 or 48 hpf. Brightfield images were acquired using a Leica M205FA fluorescence stereoscope equipped with a Leica DFC500 digital camera. For immu-nofluorescence imaging, bright-field images of embryos were obtained with a Leica DM4500B compound microscope equipped with a 20 x/0.12 NA water-immersion objectives and a QImaging Retiga-SRV digital camera. Fluorescence images were obtained with the DM4500B/Retiga-SRV system equipped with a mercury lamp and GFP (Ex: 470 nm, 40 nm bandpass; Em: 525 nm, 50 nm bandpass) filter sets.

## Western blot analysis of *Pou3f2* in zebrafish

The embryos were de-yolked in TM1 buffer (100 mM NaCl, 5 mM KCl, 5 mM HEPES pH 7.0, 1% (w/v) PEG-200,000). Twenty de-yolked embryos from each experimental condition were homogenized in SDS-PAGE loading buffer (50 μH 7.0, 1% (w/mM 2-mercaptoethanol, 4% (w/v) glycerol, 100 mM DTT, 100 mM Tris-HCl, pH 6.8), vortexed, and heated to 95°C for 5 min. The resulting lysates were used for gel electrophoresis followed by blotting with *Pou3f2* antibody (rabbit polyclonal, Abcam 137469). $\beta$-Tubulin (rabbit polyclonal, Abcam 6046) was used as loading control.

## In situ hybridization in zebrafish

The preparation of sense (used as control) and antisense RNA probes for *Kdr* and *Fli1* and in situ hybridization procedure were performed according to Thisse and Thisse (*Thisse and Thisse, 2008*).

## FACS-based analysis of GFP+ cell from zebrafish

Cells were isolated according to Shestopalov et al. (*Shestopalov et al., 2012*) with modifications. Briefly, Tg(*fli1:EGFP*)[y1] embryos at the appropriate developmental stage were dechorionated, transferred in an eppendorf tube with calcium-free Ringer's solution (200 μl for 25–30 embryos; 116 mM NaCl, 2.6 mM KCl, 5 mM HEPES, pH 7.0) and dissociated with a 200 μl pipette tip. Then 1 ml solution of 1X PBS containing trypsin (0.25%, Gibco), 50 μg collagenase P (Roche, Indianapolis, IN) and 1 mM EDTA was added and samples were incubated for 30 min at 28.5°C with further pipetting every 5 min. Enzymatic processing was quenched with stop solution (200 μl; 1X PBS containing 30% (v/v) calf serum and 10 mM CaCl2), and cells were collected by centrifugation (400 g, 5 min, 4°C). After aspirating the supernatant, cells were resuspended in a chilled solution of DMEM containing 1% (v/v) calf serum, 0.8 mM CaCl2, 50 V ml$^{-1}$ penicillin/streptomycin, centrifuged and resuspended in the same medium. The cell suspension was filtered through a 40 μm cell strainer (BD Biosciences) into FACS sample tubes. Cell suspensions were analyzed using a BD FACSAria. Wild-type zebrafish (Wik) was used as GFP negative control. Viable fluorescent single cells were identified as DAPI-negative. Cells viability was confirmed under fluorescence stereomicroscope (Leica M205) by using a Neuba-uer chamber.

## Data access

All RNA-Seq data have been deposited to the GEO database by the accession number GSE84558.

## Statistical analyses

Statistical analysis was performed with SPSS software (SPSS Inc., Chicago, IL, USA). Results were expressed as mean ± SEM. The Shapiro-Wilk test was used to confirm the null hypothesis that the data follow a normal distribution. Statistical comparisons were performed via Student t-test for two groups and via one-way ANOVA test for multiple groups. Bonferroni corrections test was applied for multiple comparisons. $p < 0.05$ was considered significant.

## Acknowledgements

This work was supported by grants to Dr. Cooke from National Institutes of Health (U01HL100397, RC2HL103400) and Cancer Prevention and Research Institute of Texas (CPRIT). Dr. Wong was

supported by a NIH PCBC Jump Start Award (PCBC_JS_2012/1_02) and American Heart Association Scientist Development Grant (AHASDG) (13SDG15800004).

## Additional information

### Funding

| Funder | Grant reference number | Author |
|--------|------------------------|--------|
| American Heart Association | 13SDG15800004 | Wing Tak Wong |
| National Institutes of Health | PCBC_JS_2012/1_02 | Wing Tak Wong |
| National Institutes of Health | U01HL100397 | John P Cooke |
| Cancer Prevention and Research Institute of Texas | | John P Cooke |
| National Institutes of Health | RC2HL103400 | John P Cooke |

The funders had no role in study design, data collection and interpretation, or the decision to submit the work for publication.

### Author contributions

WTW, Formal analysis, Investigation, Methodology, Writing—original draft; GM, Data curation, Formal analysis, Investigation, Writing—original draft, Writing—review and editing; XYT, Formal analysis, Investigation; SAT, Software, Methodology; KFA, Resources, Data curation, Software; SM, DS, DMB, Investigation, Methodology; SY, KC, Resources, Methodology; JKC, Conceptualization, Resources; HMB, Conceptualization, Resources, Methodology; JPC, Conceptualization, Resources, Supervision, Funding acquisition, Writing—review and editing

### Author ORCIDs

Gianfranco Matrone, http://orcid.org/0000-0002-6064-0734
John P Cooke, http://orcid.org/0000-0003-0033-9138

### Ethics

Animal experimentation: Zebrafish are kept according to the laboratory protocols described in "Zebrafish: A Practical Approach" (Oxford University Press, 2002). These protocols comply with the Guide for the Care and Use of Laboratory Animals, the American Association for the Accreditation of Laboratory Animal Care (AAALAC) standards, and the regulations set forth in the Animals Welfare Act (P.L. 89-544, as amended by P.L. 91-579 and P.L. 94-279). Veterinary care is provided on a 24 hours basis, including weekends and holidays, by a staff of veterinarians with specialties in laboratory animal medicine and anesthesiology, and licensed animal health technicians. Training classes are offered. All veterinary care is provided by Houston Methodist Research Institute, which is fully accredited by AAALAC (ID A4555-01) and holds an approved NIH Assurance and USDA License (start date 03/08/2013). Support includes quarantine rooms, sterile operating rooms, post-surgical recovery rooms, radiology and diagnostic laboratory services. All surgery procedures were performed under anesthesia with Tricaine 0.02 mg/ml.

## Additional files

### Major datasets

The following dataset was generated:

| Author(s) | Year | Dataset title | Dataset URL | Database, license, and accessibility information |
|-----------|------|---------------|-------------|--------------------------------------------------|
| Wing Tak Wong, Gianfranco Matrone, XiaoYu Tian | 2017 | Discovery of Novel Determinants of Endothelial Lineage: Insights from Chimeric Heterokaryons | https://www.ncbi.nlm.nih.gov/geo/query/acc.cgi?acc=GSE84558 | Publicly available at the NCBI Gene Expression Omnibus (accession no: |

GSE84558)

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
