## [Decision Letter]

Thank you for submitting your article "Discovery of Novel Determinants of Endothelial Lineage using Chimeric Heterokaryons" for consideration by *eLife*. Your article has been favorably evaluated by K VijayRaghavan (Senior Editor) and three reviewers, one of whom, Gordana Vunjak-Novakovic (Reviewer #1), is a member of our Board of Reviewing Editors. The following individual involved in review of your submission has agreed to reveal their identity: Daniel Garry (Reviewer #3).

The reviewers have discussed the reviews with one another and the Reviewing Editor has drafted this decision to help you prepare a revised submission.

Summary:

The reviewers find that the paper describes an excellent study performed by well-known experts in the field establishing and validating chimeric heterokaryons as a novel approach to identify genes involved in endothelial development and differentiation. The authors use a wide array of state-of-the-art methods including RNA seq, ChIP, endothelial functional assays in murine and human cells and the zebrafish as in vivo model. They identified novel genes involved in endothelial differentiation and focused on the role of transcription factor POU3F2 as a novel master switch orchestrating endothelial development and differentiation.

Essential revisions:

1) The factors released from endothelial cells controlling POU3F2 expression and activity in the chimeric heterokaryon model and mouse models should be discussed in more detail.

2) Please also discuss what is known about the regulation of POU3F2 in transgenic models of the Notch pathway and the Ephrin/Eph system, as POU3F2 seems to play a central role in arterio-venous differentiation. Also, provide a short discussion of potential upstream regulators of Pou3f2.

3) As many transcripts are commonly expressed in the endothelial and the hematopoietic lineages, it will be important to define the expression pattern for Pou3f2 in both the hematopoietic and endothelial lineages in either the mouse or the zebrafish embryos. Immunohistochemical and/or transcript expression (in situ hybridization) should be provided to demonstrate and define Pou3f2 expression in the developing mouse or zebrafish embryos.

4) More detailed characterization of the lethal phenotype of the Pouf2 knockdown (morpholino studies) in the zebrafish should be provided. Also, please specify if they are involved primarily in endothelial, hematopoietic or cardiac perturbations. Typically, more than one morpholino is evaluated to rule out off-target effects (in addition to the scrambled morpholino controls). Please specify if more than one morpholino has been used.

5) It would be helpful to provide one additional (conventional) technique (such as a dose response transcriptional assay and mutagenesis) to verify at least one of the downstream targets of Pou3f2. Immunohistochemical and/or transcript expression (in situ hybridization) should be included to demonstrate (and define) Pou3f2 expression in the developing mouse or zebrafish embryos.

6) Statistics should be indicated in the bar graphs of Figure 3 and Figure 5.

*Reviewer #3:*

Overall this is a well-written and well organized manuscript that identifies Pou3f2 as an important transcriptional regulator of the endothelial lineage. Strategies and data obtained (as outlined in the manuscript) will be a benefit to the field. The heterokaryon strategy is interesting and as shown by the authors it was effective in identifying endothelial transcriptional activators. Although this is an excellent manuscript, additional experiments should be included to further enhance the impact of the discovery.

4) Typically more than one morpholino is evaluated to rule out off-target effects (in addition to the scrambled morpholino controls) – did the authors use more than one morpholino in their studies?

5) Provide a short discussion of potential upstream regulators of Pou3f2 in the Discussion section.

6) Provide one additional (conventional) technique (such as a dose response transcriptional assay and mutagenesis) to verify at least one of the downstream targets of Pou3f2.

---

## [Author Response]

*Essential revisions:*

*1) The factors released from endothelial cells controlling POU3F2 expression and activity in the chimeric heterokaryon model and mouse models should be discussed in more detail.*

We now discuss potential molecules released from endothelial cells controlling Pou3f2 expression at the end of the subsection “Validation of Pou3f2 in differentiation to EC lineage”.

*2) Please also discuss what is known about the regulation of POU3F2 in transgenic models of the Notch pathway and the Ephrin/Eph system, as POU3F2 seems to play a central role in arterio-venous differentiation. Also, provide a short discussion of potential upstream regulators of Pou3f2.*

We now discuss the interaction between Pou3f2 and Notch in the first paragraph of the subsection “Importance of Pou3f2 in EC development”. The interactions amongst Pou3f2, Notch and Eph-Ephrin signaling are now addressed in the last paragraph of the aforementioned subsection.

*3) As many transcripts are commonly expressed in the endothelial and the hematopoietic lineages, it will be important to define the expression pattern for Pou3f2 in both the hematopoietic and endothelial lineages in either the mouse or the zebrafish embryos. Immunohistochemical and/or transcript expression (in situ hybridization) should be provided to demonstrate and define Pou3f2 expression in the developing mouse or zebrafish embryos.*

This is an excellent suggestion. Accordingly we now show our work to address this suggestion in Figure 4—figure supplement 1 and Figure 4—figure supplement 2. In Figure 4—figure supplement 1, we show our data derived from in situ hybridization for Pou3f2 in the zebrafish embryo. In Figure 4—figure supplement 2, we show our work to define the expression pattern for Pou3f2 in both the hematopoietic and endothelial lineages. We performed Real time PCR in FACS-purified GFP+ cells isolated from *tg(fli1:EGFP)* or *tg(cmyb:GFP)* embryos, where Fli1 and Cmyb are respectively endothelial and hematopoietic lineage markers. In Figure 4—figure supplement 2 the levels of expression of Pou3f2 in these two types of cells are shown. Both Figure 4—figure supplement 1 and Figure 4—figure supplement 2 are discussed in the last paragraph of the subsection 2 Importance of Pou3f2 in EC development”.

*4) More detailed characterization of the lethal phenotype of the Pouf2 knockdown (morpholino studies) in the zebrafish should be provided. Also, please specify if they are involved primarily in endothelial, hematopoietic or cardiac perturbations. Typically, more than one morpholino is evaluated to rule out off-target effects (in addition to the scrambled morpholino controls). Please specify if more than one morpholino has been used.*

Discussion of the lethal phenotype of the Pou3f2 KD is now provided in the last paragraph of the subsection “Importance of Pou3f2 in EC development”. We also clarified that two Pou3f2-targeted morpholinos were used in the KD studies. More detailed information and the sequences of the Morpholinos are now provided in the subsection “Morpholinos and Caged Morpholinos injections”.

To further address this comment, we also injected Pou3f2 morpholino in *tg(cmyb:EGFP)* and showed that Pou3f2 KD did not alter the percentage of c-myb expressing cells. These new data are now discussed in the last paragraph of the subsection “Importance of Pou3f2 in EC development”.

*5) It would be helpful to provide one additional (conventional) technique (such as a dose response transcriptional assay and mutagenesis) to verify at least one of the downstream targets of Pou3f2. Immunohistochemical and/or transcript expression (in situ hybridization) should be included to demonstrate (and define) Pou3f2 expression in the developing mouse or zebrafish embryos.*

In our new Figure 4—figure supplement 3 we now show real time PCR data in the zebrafish embryos following Pou3f2 KD showing significant inhibition of Fli1 and Kdr expression, confirming our previous in-situ hybridization data reported in Figure 4.

In Figure 4—figure supplement 1 we report in situ hybridization in the zebrafish embryo specific for Pou3f2 demonstrating the expression and localization of this transcription factor.

6) Statistics should be indicated in the bar graphs of Figure 3 and Figure 5.

We now provide these analyses.

*Reviewer #3:*

*Overall this is a well-written and well organized manuscript that identifies Pou3f2 as an important transcriptional regulator of the endothelial lineage. Strategies and data obtained (as outlined in the manuscript) will be a benefit to the field. The heterokaryon strategy is interesting and as shown by the authors it was effective in identifying endothelial transcriptional activators. Although this is an excellent manuscript, additional experiments should be included to further enhance the impact of the discovery.*

*4) Typically more than one morpholino is evaluated to rule out off-target effects (in addition to the scrambled morpholino controls) – did the authors use more than one morpholino in their studies?*

We have now clarified that two Pou3f2-targeted morpholino were used in KD studies. More detailed information and the sequences of the morpholinos are now provided in the subsection “Morpholinos and Caged Morpholinos injections”.

*5) Provide a short discussion of potential upstream regulators of Pou3f2 in the Discussion section.*

We now provide this discussion in the last paragraph of the subsection “Validation of Pou3f2 in differentiation to EC lineage” and in the last paragraph of the subsection “Importance of Pou3f2 in EC development”.

6) Provide one additional (conventional) technique (such as a dose response transcriptional assay and mutagenesis) to verify at least one of the downstream targets of Pou3f2

In Figure 4—figure supplement 3 we now include real time PCR data in the zebrafish embryos following Pou3f2 KD showing significant inhibition of Fli1 and Kdr expression, confirming our previous in-situ hybridization data reported in Figure 4.